# ImplicitSLIM and How it Improves Embedding-Based Collaborative Filtering

**Ilya Shenbin & Sergey Nikolenko**
St. Petersburg Department of the Steklov Mathematical Institute of the RAS
St. Petersburg, Russia
`ilya.shenbin@gmail.com, sergey@logic.pdmi.ras.ru`

## Abstract

We present ImplicitSLIM, a novel unsupervised learning approach for sparse high-dimensional data, with applications to collaborative filtering. Sparse linear methods (SLIM) and their variations show outstanding performance, but they are memory-intensive and hard to scale. ImplicitSLIM improves embedding-based models by extracting embeddings from SLIM-like models in a computationally cheap and memory-efficient way, without explicit learning of heavy SLIM-like models. We show that ImplicitSLIM improves performance and speeds up convergence for both state of the art and classical collaborative filtering methods. The source code for ImplicitSLIM, related models, and applications is available at `https://github.com/ilya-shenbin/ImplicitSLIM`.

## 1 Introduction

Learnable embeddings are a core part of many collaborative filtering (CF) models. Often they are introduced and learned explicitly: e.g., matrix factorization (MF) models express feedback as a product of two embedding matrices (Mnih & Salakhutdinov, 2007; Bell et al., 2007), autoencoders map user feedback to embeddings via neural networks or other parameterized functions (Sedhain et al., 2015; Liang et al., 2018; Lobel et al., 2019; Shenbin et al., 2020; Mirvakhabova et al., 2020), while graph convolutional networks (GCN) (Wang et al., 2019; He et al., 2020) perform a diffusion process between user and item embeddings starting from some initial values. In this work, we propose an approach able to improve a wide variety of collaborative filtering models with learnable embeddings.

Item-item methods, including kNN-based approaches (Sarwar et al., 2001) and sparse linear methods (SLIM) (Ning & Karypis, 2011), are making predictions based on item-item similarity. Previous research shows that the item-item weight matrix learned by SLIM-like models can become a part of other collaborative filtering models; e.g., RecWalk uses it as a transition probability matrix (Nikolakopoulos & Karypis, 2019). In this work, we reuse the item-item weight matrix in order to enrich embedding-based models with information on item-item interactions. Another motivation for our approach stems from nonlinear dimensionality reduction methods (e.g., VAEs) applied to collaborative filtering (Shenbin et al., 2020). We consider a group of *manifold learning* methods that aim to preserve the structure of data in the embedding space, that is, they force embeddings of similar objects to be similar. A related idea has already been used in collaborative filtering models, e.g. by Rao et al. (2015).

We propose an approach called *ImplicitSLIM* that performs nonlinear unsupervised learning of embeddings that can be further used for making recommendations. The name *ImplicitSLIM* means that it allows to extract useful data from SLIM-like models *without explicitly training SLIM itself* and does not refer to models with explicit or implicit feedback (however, we consider only the case of implicit feedback). *ImplicitSLIM* learns item or user embeddings with closed form solutions that do not require computationally intensive operations. The resulting embeddings can be used for initialization or regularization of item and/or user embeddings for a wide variety of other methods, from matrix factorization to autoencoder-based approaches and models based on graph convolutional networks, and also for RNN- or Transformer-based sequential models. In most cases, we find that *ImplicitSLIM* improves the final recommendation results, often very significantly. By adding *Im-*

*plicitSLIM* to nonlinear VAEs such as RecVAE (Shenbin et al., 2020) and H+Vamp(Gated) (Kim & Suh, 2019), we have been able to improve state of the art results on the *MovieLens-20M* and *Netflix Prize* datasets.

Our method builds upon two known approaches. First, we consider locally linear embeddings (LLE) (Roweis & Saul, 2000), a nonlinear manifold learning method which is surprisingly well suited for our idea. LLE consists of two linear steps. The first step is very similar to SLIM, that is, it learns the weights for optimal reconstruction of data samples from their nearest neighbours, and on the second step low-dimensional embeddings are extracted from learned weights. In *ImplicitSLIM*, we modify this procedure as follows: first, extracted embeddings are obtained given some prior embeddings, that is, we turn LLE into a procedure that consistently improves embeddings; second, we compress the two-step procedure into a single step without explicit computation of the intermediate weight matrix, which significantly reduces the costs in both computation and memory. Another approach that we use in our method is EASE (Steck, 2019). It is a slightly simplified case of SLIM with a closed form solution. A later empirical study by Steck et al. (2020) showed that features of SLIM that are dropped in EASE do not significantly improve performance in case of a large number of users. In this work, we consider *ImplicitSLIM* based only on EASE, but we note upfront that the proposed approach could also be based on some other variations of SLIM with a closed-form solution, e.g., full-rank DLAE or EDLAE (Steck, 2020). Although the proposed method learns low-dimensional embeddings, we do not aim to construct a low-rank approximation of SLIM, as do, e.g., Jin et al. (2021) and Kabbur et al. (2013), but aim to improve other existing models with SLIM.

Below, Section 2 shows the necessary background, Section 3 develops *ImplicitSLIM*, Section 4 integrates it into various collaborative filtering models, Section 5 introduces the baselines and presents our experimental evaluation, Section 7 reviews related work, and Section 8 concludes the paper.

## 2 PRELIMINARIES

In this section we describe the approaches that provide the background and motivation for *ImplicitSLIM*; for a survey of other related work see Section 7. Consider an implicit feedback matrix $\mathbf{X} \in \{0, 1\}^{U \times I}$, where $\mathbf{X}_{ui} = 1$ if the $u$-th user positively interacted with (liked, watched etc.) the $i$-th item (movie, good etc.) and $\mathbf{X}_{ui} = 0$ otherwise, and $U$ and $I$ are the numbers of users and items respectively. For low-rank models, we denote the embedding dimension by $L$.

**Sparse Linear Methods** (SLIM) is a family of item-based approaches originally proposed by Ning & Karypis (2011) and further extended by Christakopoulou & Karypis (2014; 2016); Steck et al. (2020), and Steck (2020). The core idea of SLIM is to learn to reconstruct user feedback for a given item as a linear combination of feedback from the same user for other items. We focus on *Embarrassingly Shallow Autoencoders* (EASE) (Steck, 2019), a special case of SLIM that, critically for our approach, admits a closed form solution. EASE can be trained by solving the following optimization task:

$$\hat{\mathbf{B}} = \operatorname{argmin}_{\mathbf{B}} \ \|\mathbf{X} - \mathbf{X}\mathbf{B}\|_F^2 + \lambda\|\mathbf{B}\|_F^2 \qquad \text{s.t.} \quad \operatorname{diag} \mathbf{B} = \mathbf{0}, \tag{1}$$

where $\|\cdot\|_F$ is the Frobenius norm and $\operatorname{diag} \mathbf{B}$ is the diagonal of $\mathbf{B}$; the constraint forbids the trivial solution $\hat{\mathbf{B}} = \mathbf{I}$. Problem (1) can be solved exactly with Lagrange multipliers, getting

$$\hat{\mathbf{B}} = \mathbf{I} - \hat{\mathbf{P}} \operatorname{diagMat}(\mathbf{1} \oslash \operatorname{diag} \hat{\mathbf{P}}), \tag{2}$$

where $\hat{\mathbf{P}} \stackrel{\text{def}}{=} (\mathbf{X}^\top\mathbf{X} + \lambda\mathbf{I})^{-1}$, $\operatorname{diagMat}(\mathbf{x})$ is a diagonal matrix with vector $\mathbf{x}$ on the diagonal, $\mathbf{1}$ is a vector of ones, and $\oslash$ denotes element-wise division; see Steck (2019) for detailed derivations. SLIM has impressive performance but quadratic space complexity and nearly cubic time complexity. Elements of $\hat{\mathbf{B}}$ that are close to zero can be zeroed, which leads to a highly sparse matrix and can partially solve the memory issue, but it will not reduce memory costs during training.

**Locally Linear Embeddings** (LLE) is a nonlinear dimensionality reduction method (Roweis & Saul, 2000; Saul & Roweis, 2003; Chojnacki & Brooks, 2009; Ghojogh et al., 2020). For consistency, we introduce LLE in modified notation; Appendix A.1 links it to the original. We denote the $i$-th data sample as $\mathbf{X}_{*i}$ and the set of indices for nearest neighbors of the $i$-th sample by $\operatorname{NN}(i)$. LLE works in two steps. First, it finds the matrix of local coordinate vectors $\hat{\mathbf{B}}$ as

$$\hat{\mathbf{B}} = \operatorname{argmin}_{\mathbf{B}} \ \sum_i \|\mathbf{X}_{*i} - \sum_{j \in \operatorname{NN}(i)} \mathbf{X}_{*j}\mathbf{B}_{ji}\|_2^2 \quad \text{s.t.} \quad \mathbf{B}^\top\mathbf{1} = \mathbf{1}, \tag{3}$$

where $\hat{\mathbf{B}}_{ji} = 0$ if $j \notin \mathrm{NN}(i)$. The matrix $\hat{\mathbf{B}}$ is invariant to scaling, orthogonal transformations, and translations of data samples; note that translation invariance is a result of the sum-to-one constraint.

On the second step, LLE finds the embedding matrix $\hat{\mathbf{V}}$ given the sparse matrix $\hat{\mathbf{B}}$:

$$\hat{\mathbf{V}} = \mathrm{argmin}_{\mathbf{V}} \ \|\mathbf{V} - \mathbf{V}\hat{\mathbf{B}}\|_F^2 \quad \text{s.t.} \quad \mathbf{V}\mathbf{V}^\top = n\mathbf{I}, \quad \mathbf{V}\mathbf{1} = \mathbf{0}. \tag{4}$$

The first constraint forbids the zero solution; the second removes the translational degree of freedom.

**Regularization of embeddings** plays an important role for collaborative filtering models, could be performed with uninformative priors on embeddings (Mnih & Salakhutdinov, 2007), context information (McAuley & Leskovec, 2013; Ling et al., 2014), or information on item-item interactions to regularize item embeddings (or, symmetrically, user embeddings). Liang et al. (2016) and Nguyen et al. (2017) learn matrix factorization embeddings jointly with item embeddings obtained from the cooccurrence matrix with a *word2vec*-like approach. Rao et al. (2015) use graph regularization, minimizing the distance between item embeddings that are similar according to a given adjacency matrix $\tilde{\mathbf{A}}$, i.e., optimizing the following penalty function:

$$\mathcal{L}_{\mathrm{GRAPH\,REG}}(\mathbf{Q}) = \mathrm{tr}\left(\mathbf{Q}\mathbf{L}\mathbf{Q}^\top\right) = \sum_{i,j} \tilde{\mathbf{A}}_{ij}\|\mathbf{Q}_{*i} - \mathbf{Q}_{*j}\|_2^2, \tag{5}$$

where $\mathbf{L}$ is the graph Laplacian matrix corresponding to adjacency matrix $\tilde{\mathbf{A}}$ and $\mathbf{Q}$ is the $L \times I$ item embedding matrix. We will return to regularization of item embeddings in Section 4.

## 3 PROPOSED APPROACH

### 3.1 SLIM AS LLE

Consider the first step of LLE defined in (3). We propose to use the neighborhood function $\mathrm{NN}(i) = \{1, 2, \ldots, I\}\backslash\{i\}$; we can rewrite (3) as

$$\hat{\mathbf{B}} = \mathrm{argmin}_{\mathbf{B}} \ \|\mathbf{X} - \mathbf{X}\mathbf{B}\|_F^2, \quad \text{s.t.} \quad \mathbf{B}^\top\mathbf{1} = \mathbf{1}, \quad \mathrm{diag}\,\mathbf{B} = \mathbf{0}.$$

Due to this choice of the neighborhood function, we obtain a simpler closed-form solution for this optimization problem (compared to the general solution from LLE), which will make it possible to perform the main trick in Section 3.2. Moreover, now this problem is very similar to the EASE optimization task (1), but with an additional constraint that columns of $\mathbf{B}$ sum to one and without a Frobenius norm regularizer on $\mathbf{B}$. The latter was claimed to be an important feature of SLIM (Steck, 2019); a similar regularizer was proposed by the authors of LLE (Saul & Roweis, 2003). Therefore, we propose to bring it back, computing $\hat{\mathbf{B}}$ as follows:

$$\hat{\mathbf{B}} = \mathrm{argmin}_{\mathbf{B}} \ \|\mathbf{X} - \mathbf{X}\mathbf{B}\|_F^2 + \lambda\|\mathbf{B}\|_F^2 \quad \text{s.t.} \ \mathbf{B}^\top\mathbf{1} = \mathbf{1}, \ \mathrm{diag}\,\mathbf{B} = \mathbf{0}. \tag{6}$$

See Appendix A.2 for the solution of this optimization task. Thus, we have introduced a new form of LLE and shown that EASE can be considered as the first step of LLE without the sum-to-one constraint. Now we can extract item embeddings with LLE using a first step inspired by EASE: (i) first compute $\hat{\mathbf{B}}$ with (6) using EASE with the sum-to-one constraint, then (ii) find the embeddings by solving (4). We call this approach SLIM-LLE. The same procedure with the transposed feedback matrix $\mathbf{X}^\top$ will extract user embeddings instead of item embeddings. However, SLIM-LLE is computationally intensive, not memory efficient, and cannot use available embeddings as priors. In what follows, we fix these drawbacks.

### 3.2 IMPLICITSLIM

We now revisit the second step of LLE (4). Suppose that we want to obtain embeddings close to a given matrix of item embeddings $\mathbf{Q}$. Instead of (4), we introduce the following unconstrained optimization problem:

$$\hat{\mathbf{V}} = \mathrm{argmin}_{\mathbf{V}} \ \|\mathbf{V} - \mathbf{V}\hat{\mathbf{B}}\|_F^2 + \alpha\|(\mathbf{V} - \mathbf{Q})\mathbf{A}^\top\|_F^2, \tag{7}$$

where $\alpha$ is a nonnegative regularization coefficient and $\mathbf{A}$ is an auxiliary weight matrix whose key purpose will be exposed at the end of this subsection. As we will see below, this problem has a

unique nonzero solution, hence constraints from the second step of LLE are not necessary here. The sum-to-one constraint in the first step of LLE (3) implicitly ensures the sum-to-zero constraint of the second step of LLE (4), but we have redefined the second step (7) as an unconstrained optimization problem, so now there is no technical need in the sum-to-one constraint; it also provides translational invariance, but we do not have a good reason why it could be useful here. Hence we drop the sum-to-one constraint to obtain a simpler closed form solution for the first step, i.e., we use the EASE problem (1) directly as the first step of LLE (Appendix E.2 examines this choice empirically). The ultimate reason for dropping the sum-to-one constraint is that the resulting form of the first step will allow us to avoid explicitly calculating the matrix $\hat{\mathbf{B}}$, as we will show below in this section.

We can now find a closed form solution for (7):

$$\hat{\mathbf{V}} = \alpha \mathbf{Q} \mathbf{A}^\top \mathbf{A} \left( (\hat{\mathbf{B}} - \mathbf{I})(\hat{\mathbf{B}} - \mathbf{I})^\top + \alpha \mathbf{A}^\top \mathbf{A} \right)^{-1}. \tag{8}$$

Now we can substitute here the first step solution (2), which further (drastically) simplifies the calculations. Recall from (2) that $\hat{\mathbf{B}} = \mathbf{I} - \hat{\mathbf{P}} \mathbf{D}_{\hat{\mathbf{P}}}^{-1}$, where $\mathbf{D}_{\hat{\mathbf{P}}} \stackrel{\text{def}}{=} \mathrm{diagMat}(\mathrm{diag}\,\hat{\mathbf{P}})$, i.e., $\mathbf{D}_{\hat{\mathbf{P}}}$ is equal to $\hat{\mathbf{P}}$ with zeroed non-diagonal elements. Also recall that $\hat{\mathbf{P}} \stackrel{\text{def}}{=} (\mathbf{X}^\top \mathbf{X} + \lambda \mathbf{I})^{-1}$, where $\lambda$ was introduced in (1), so $\hat{\mathbf{P}}$ is symmetric. Therefore,

$$\hat{\mathbf{V}} = \alpha \mathbf{Q} \mathbf{A}^\top \mathbf{A} \left( \hat{\mathbf{P}} \mathbf{D}_{\hat{\mathbf{P}}}^{-2} \hat{\mathbf{P}} + \alpha \mathbf{A}^\top \mathbf{A} \right)^{-1}. \tag{9}$$

We now see that (7) actually has a unique nonzero solution if $\hat{\mathbf{P}}$ is full-rank, which is true if $\lambda$ from (1) is positive. To compute it, we have to invert $I \times I$ matrices twice, which is the main computational bottleneck here. Using the Woodbury matrix identity, we rewrite (9) as

$$\hat{\mathbf{V}} = \alpha \mathbf{Q} \mathbf{A}^\top \mathbf{A} \left( \mathbf{R}^{-1} - \mathbf{R}^{-1} \mathbf{A}^\top \left( \alpha^{-1} \mathbf{I} + \mathbf{A} \mathbf{R}^{-1} \mathbf{A}^\top \right)^{-1} \mathbf{A} \mathbf{R}^{-1} \right), \tag{10}$$

where $\mathbf{R}$ is introduced to abbreviate the formulas and is defined as

$$\mathbf{R} \stackrel{\text{def}}{=} \hat{\mathbf{P}} \mathbf{D}_{\hat{\mathbf{P}}}^{-2} \hat{\mathbf{P}}, \quad \text{i.e.,} \quad \mathbf{R}^{-1} = (\mathbf{X}^\top \mathbf{X} + \lambda \mathbf{I}) \mathbf{D}_{\hat{\mathbf{P}}}^2 (\mathbf{X}^\top \mathbf{X} + \lambda \mathbf{I}). \tag{11}$$

The most troublesome multiplier here is $\mathbf{D}_{\hat{\mathbf{P}}}$; we approximate it as (see Appendix A.3 for derivations)

$$\mathbf{D}_{\hat{\mathbf{P}}} \approx \mathbf{D}_{\hat{\mathbf{P}}^{-1}}^{-1} = \mathrm{diagMat}(\mathbf{1} \oslash \mathrm{diag}(\mathbf{X}^\top \mathbf{X} + \lambda \mathbf{I})). \tag{12}$$

Assuming $\mathbf{A}$ is an $L \times I$ matrix with $L \ll I$, and using the approximation shown above, we avoid the inversion of $I \times I$ matrices in (10) by reducing it to inverting $L \times L$ matrices. Moreover, we can avoid even storing and multiplying $I \times I$ matrices with a closer look at (10). First, we expand the brackets in (10) and (11), and calculate $\mathbf{A} \mathbf{R}^{-1}$, which we denote as $\mathbf{F} \stackrel{\text{def}}{=} \mathbf{A} \mathbf{R}^{-1}$:

$$\mathbf{F} = \mathbf{A} \mathbf{X}^\top \mathbf{X} \mathbf{D}_{\hat{\mathbf{P}}}^2 \mathbf{X}^\top \mathbf{X} + \lambda \mathbf{A} \mathbf{D}_{\hat{\mathbf{P}}}^2 \mathbf{X}^\top \mathbf{X} + \lambda \mathbf{A} \mathbf{X}^\top \mathbf{X} \mathbf{D}_{\hat{\mathbf{P}}}^2 + \lambda^2 \mathbf{A} \mathbf{D}_{\hat{\mathbf{P}}}^2. \tag{13}$$

Now we can rewrite (10) as

$$\begin{aligned} \hat{\mathbf{V}} &= \alpha \mathbf{Q} \mathbf{A}^\top (\mathbf{F} - \mathbf{F} \mathbf{A}^\top (\alpha^{-1} \mathbf{I} + \mathbf{F} \mathbf{A}^\top)^{-1} \mathbf{F}) = \\ &= \alpha \mathbf{Q} \mathbf{A}^\top (\mathbf{F} - (\mathbf{I} - (\mathbf{I} + \alpha \mathbf{F} \mathbf{A}^\top)^{-1}) \mathbf{F}). \end{aligned} \tag{14}$$

In (13) and (14), any intermediate result is at most an $L \times \max(U, I)$ matrix, which makes *ImplicitSLIM* much more computationally and memory efficient than explicit computation of (2) and (8). Now we are dealing only with matrices of moderate size, whose number of elements depends linearly on the number of items or users in the worst case, except for the sparse feedback matrix $\mathbf{X}$. Note that we have introduced the auxiliary weight matrix $\mathbf{A}$ in (7) and then assumed that it is an $L \times I$ matrix with $L \ll I$, but we still have not specified the matrix $\mathbf{A}$. We propose to set it equal to $\mathbf{Q}$, so the regularizer $\|\mathbf{V}\mathbf{Q}^\top - \mathbf{Q}\mathbf{Q}^\top\|_F^2$ from (7) could be considered as approximate pairwise distance. Appendix E.2 compares it empirically with a more natural-looking regularizer $\|\mathbf{V} - \mathbf{Q}\|_F^2$.

Overall, in this section we have motivated and introduced *ImplicitSLIM*, a method inspired by SLIM and LLE; we have shown how it can be derived and efficiently implemented. We also showed a predecessor of *ImplicitSLIM*, SLIM-LLE, which is a special case of LLE.

# 4 IMPLICITSLIM WITH OTHER MODELS

## 4.1 GENERAL SCENARIO

*ImplicitSLIM* is not a standalone approach. Below, we show how embeddings obtained with *ImplicitSLIM* can be applied to existing models. First, the item embeddings matrix $\mathbf{Q}$ of a given model can be initialized with *ImplicitSLIM* or SLIM-LLE. Since *ImplicitSLIM* itself requires an embedding matrix as input, which could be initialized either randomly (e.g., from the standard normal distribution), with the output of *ImplicitSLIM*/SLIM-LLE (improving the embeddings iteratively), or with an external model. Moreover, when we are training a collaborative filtering model we can send its embeddings matrix $\mathbf{Q}$ to *ImplicitSLIM* and replace it with the result ($\hat{\mathbf{V}}$ in Section 3.2) immediately before the update of $\mathbf{Q}$ (not necessarily *every* update). This procedure may be less stable than minimizing the distance between $\mathbf{Q}$ and *ImplicitSLIM* output but requires fewer calls to *ImplicitSLIM*.

## 4.2 SLIM REGULARIZATION

In this section we first briefly suspend the discussion of *ImplicitSLIM* to introduce and motivate the *SLIM regularizer*, and then show how we can take advantage of it in an efficient way via *ImplicitSLIM*. Inspired by LLE, specifically by its second step (4), we define the *SLIM regularizer* as

$$\mathcal{L}_{\text{SLIM REG}}(\mathbf{Q}) = \|\mathbf{Q} - \mathbf{Q}\hat{\mathbf{B}}\|_F^2 = \sum_i \|\mathbf{Q}_{*i} - \sum_j \mathbf{Q}_{*j}\hat{\mathbf{B}}_{ji}\|_2^2, \tag{15}$$

where $\hat{\mathbf{B}}$ is the item-item similarity matrix from SLIM. This penalty function forces each item embedding to be close to the linear combination of other embeddings with coefficients learned by SLIM. Unfortunately, using this regularizer could be computationally costly and, moreover, it needs a precomputed matrix $\hat{\mathbf{B}}$, which takes away scalability. We emphasize that $\mathcal{L}_{\text{SLIM REG}}$ is equal to the graph regularizer (5) for certain matrices $\hat{\mathbf{B}}$, e.g. for one obtained from SLIM with the sum-to-one constraint (6); see Appendix A.4 for details.

Let $\mathcal{L}_{\text{CF}}(\Theta, \mathbf{Q})$ be the loss function of some collaborative filtering model, where $\mathbf{Q}$ is the embedding matrix and $\Theta$ represents other model parameters. Adding the *SLIM regularizer*, we get the following loss function:

$$\mathcal{L}_{\text{CF + SLIM REG}}(\Theta, \mathbf{Q}) = \mathcal{L}_{\text{CF}}(\Theta, \mathbf{Q}) + \mathcal{L}_{\text{SLIM REG}}(\mathbf{Q}) \tag{16}$$

We propose to optimize the resulting loss function by alternating optimization of both terms. In order to perform this optimization, we relax $\mathcal{L}_{\text{CF + SLIM REG}}$ as

$$\mathcal{L}_{\text{CF + SLIM REG}}^{\text{RELAXED}}(\Theta, \mathbf{Q}, \mathbf{V}) = \mathcal{L}_{\text{CF}}(\Theta, \mathbf{Q}) + \mathcal{L}_{\text{SLIM REG}}(\mathbf{V}) + \alpha \cdot d(\mathbf{Q}, \mathbf{V}),$$

where $d(\mathbf{Q}, \mathbf{V})$ is the distance between two embedding matrices $\mathbf{Q}$ and $\mathbf{V}$ such that $\mathcal{L}_{\text{CF + SLIM REG}}(\Theta, \mathbf{Q}) = \mathcal{L}_{\text{CF + SLIM REG}}^{\text{RELAXED}}(\Theta, \mathbf{Q}, \mathbf{Q})$. Now we can update the loss function w.r.t. $(\Theta, \mathbf{Q})$ and $\mathbf{V}$ alternately. Let $d(\mathbf{Q}, \mathbf{V}) = \|(\mathbf{V} - \mathbf{Q})\mathbf{A}^\top\|_F^2$; then

$$\hat{\mathbf{V}} = \text{argmin}_{\mathbf{V}} \ \mathcal{L}_{\text{CF + SLIM REG}}^{\text{RELAXED}}(\Theta, \mathbf{Q}, \mathbf{V}) = \text{argmin}_{\mathbf{V}} \ \|\mathbf{V} - \mathbf{V}\hat{\mathbf{B}}\|_F^2 + \alpha\|(\mathbf{V} - \mathbf{Q})\mathbf{A}^\top\|_F^2,$$

which is the same as (7), so $\hat{\mathbf{V}}$ is equal to the embeddings extracted with *ImplicitSLIM*. This means that we can compute $\hat{\mathbf{V}}$ efficiently and even without an explicit computation of $\hat{\mathbf{B}}$ by using the results of Section 3.2.

# 5 EXPERIMENTAL EVALUATION

For experiments, we follow the basic evaluation setup by Liang et al. (2018). We test the proposed approaches on three datasets: *MovieLens-20M*[1] (Harper & Konstan, 2015), *Netflix Prize Dataset*[2] (Bennett et al., 2007), and *Million Songs Dataset*[3] (Bertin-Mahieux et al., 2011), comparing models in terms of the ranking metrics *Recall@k* and *NDCG@k*. Experiments are conducted in the strong generalization setting (Marlin, 2004): users in the training, validation, and test

---

[1] https://grouplens.org/datasets/movielens/20m/

[2] https://www.netflixprize.com/

[3] http://millionsongdataset.com/

subsets are disjoint. We also perform additional benchmarking in the setup by He et al. (2020) on *Yelp2018* (Wang et al., 2019) and *MovieLens-1M*[4] (Harper & Konstan, 2015) datasets. There datasets are split into train/test subsets along the ratings, i.e., according to the weak generalization setting. Moreover, we test our approach in the setup by Sun et al. (2019) on the *MovieLens-1M* and *MovieLens-20M* datasets discussed above. Here the last and the penultimate feedback of every user are used as test and validation data respectively, while the remaining feedback is used as training data. In this setting, sampled metrics are employed in order to speed up computation. The choice of these three setups is determined by the models to which we apply the proposed method. In order to achieve a fair comparison of the results, we inherited dataset splits, metrics, and evaluation strategies from the corresponding setups.

We have selected a number of different models as baselines: (1) *matrix factorization* (MF); we consider MF trained with ALS with uniform weights (Hu et al., 2008), which is a simple and computationally efficient baseline, and also weighted matrix factorization (WMF) (Hu et al., 2008) trained with eALS (He et al., 2016); (2) MF augmented with *regularization based on item-item interactions*; here we selected *CoFactor* (Liang et al., 2016), which combines MF and *word2vec*, and GRALS (Rao et al., 2015) that employs graph regularization; (3) *linear models*; we have chosen full-rank models SLIM (Ning & Karypis, 2011) and EASE (Steck, 2019) and a low-rank model PLRec (Sedhain et al., 2016) (we use only *I-Linear-FLow*); (4) *nonlinear autoencoders*; here we consider the shallow autoencoder CDAE (Wu et al., 2016), variational autoencoder MultVAE (Liang et al., 2018), and its successors: RaCT (Lobel et al., 2019), RecVAE (Shenbin et al., 2020), and H+Vamp(Gated) (Kim & Suh, 2019). Details on the implementation and evaluation of baselines are provided in Appendix B.

*ImplicitSLIM* is not a standalone approach; thus, we have chosen several downstream models to evaluate it. As MF and PLRec are simple and lightweight models, we used them as downstream models for extended experiments with *ImplicitSLIM* and SLIM-LLE. There are lots of ways to utilize embeddings from *ImplicitSLIM* in downstream models, but for the experimental section we have selected several that are most demonstrative and effective. Apart from MF evaluation in its **vanilla** form, we use the following setups for MF as a downstream model: (1) **ImplicitSLIM init+reg**, where item embeddings are regularized by *ImplicitSLIM* and initialized by *ImplicitSLIM* that gets a sample from the standard normal distribution as input; (2) **SLIM-LLE init** and **ImplicitSLIM init**, where item embeddings are initialized respectively by SLIM-LLE and by applying *ImplicitSLIM* repeatedly several times; in both cases, the corresponding loss function is minimized to obtain user embeddings at the validation/test stage only, i.e., there is no training phase, only hyperparameter search; We use similar setups for PLRec-based experiments. PLRec has two item embedding matrices, $\mathbf{W}$ and $\mathbf{Q}$; the first one is fixed, and the latter is learnable. When using PLRec with setups defined above, initialization applies only to $\mathbf{W}$: it is meaningless to initialize $\mathbf{Q}$ since we have a closed form solution for $\mathbf{Q}$; on the other hand, regularization is applied only to $\mathbf{Q}$ since $\mathbf{W}$ is fixed. Other differences are that the **SLIM-LLE init** setup applied to PLRec has a training phase which is a single update of $\mathbf{Q}$, and in **ImplicitSLIM init+reg** it is initialized by applying *ImplicitSLIM* repeatedly.

We also have applied *ImplicitSLIM* to other methods. WMF is hard to scale, so we tested it only with small embedding dimensions in the **ImplicitSLIM init+reg** setup. Among nonlinear autoencoders we selected the most successful ones, namely RecVAE and H+Vamp(Gated), to evaluate them with *ImplicitSLIM* as regularizer. Unlike MF and PLRec, these models are trained using stochastic gradient descent (SGD), so we need a different scenario of applying *ImplicitSLIM*. Namely, we have found that the best performing procedure for models of this kind is to feed current item embeddings to *ImplicitSLIM* and update them with the resulting embeddings.

We did not apply *ImplicitSLIM* to *CoFactor* and GRALS since both have their own regularization techniques that utilize item-item interaction. We also did not apply *ImplicitSLIM* to SLIM and EASE since our approach can be applied to embedding-based models only.

In addition, we consider UltraGCN (Mao et al., 2021), which is a state of the art GCN-influenced embedding-based model according to Zhu et al. (2022). It has trainable embeddings for both users and items, and the same users are considered in both train and test time. This allows us to apply *ImplicitSLIM* not only for item embeddings but also for user embeddings. In both cases we

---

[4]https://grouplens.org/datasets/movielens/1m/

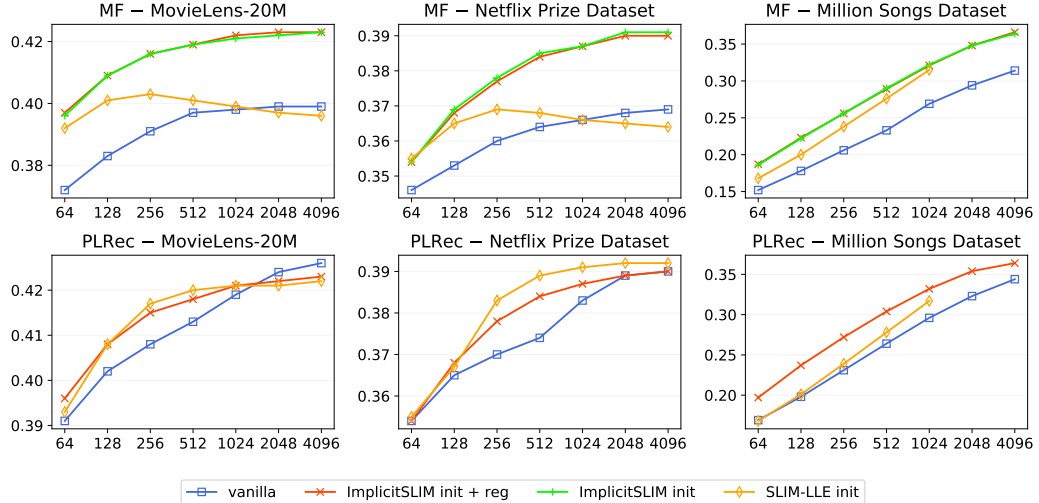

Figure 1: ImplicitSLIM and SLIM-LLE applied to MF and PLRec (setups defined in Section 5); the X-axis shows embedding dimensions, the Y-axis shows NDCG@100.

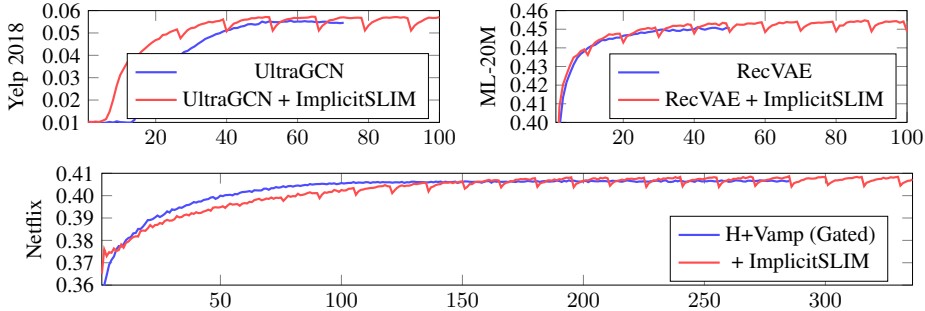

Figure 2: Sample convergence plots for state of the art models with ImpicitSLIM; the X-axis shows training epochs; Y-axis, NDCG@20 metric on Yelp2018, NDCG@100 metric on other datasets.

update embeddings once in several epochs, similarly to nonlinear autoencoders. We also consider BERT4Rec (Sun et al., 2019) as a popular sequential model with strong performance. See Appendix D for a detailed description of experimental setups and pseudocode.

## 6 RESULTS

Figure 1 presents the results of applying *ImplicitSLIM* and its variations to matrix factorization (MF) and PLRec for different embedding dimensions (X-axis). Using *ImplicitSLIM* for both initialization and regularization yields the best results in most cases. More interestingly, initialization by *ImplicitSLIM* shows nearly equivalent performance, which means that embeddings obtained with *ImplicitSLIM* are good enough to not have to update them with alternating least squares. Thus, *ImplicitSLIM* is not only more computationally efficient than other models but also performs better in terms of prediction scores. The performance of SLIM-LLE varies greatly depending on the dataset and downstream model, and in some cases outperforms *ImplicitSLIM*, but it lacks scalability (results shown for SLIM-LLE are limited in embedding dimensions due to high computational costs).

Table 1 shows that MF and PLRec regularized by *ImplicitSLIM* are approaching EASE in performance, especially in high dimensions, but are computationally cheaper and more memory-efficient in most cases. Fig. 3 shows when *ImplicitSLIM* is preferable in terms of wall clock time and memory usage compared to vanilla MF and EASE (see Appendix E.1 for running time details). Note, however, that EASE has only one hyperparameter while *ImplicitSLIM* and MF together have five.

Table 1: Experimental results. The best results are highlighted in bold.

| Model | MovieLens-20M | | | Netflix Prize Dataset | | | Million Songs Dataset | | |
|---|---|---|---|---|---|---|---|---|---|
| | Recall @20 | Recall @50 | NDCG @100 | Recall @20 | Recall @50 | NDCG @100 | Recall @20 | Recall @50 | NDCG @100 |
| **Matrix factorization** | | | | | | | | | |
| MF | 0.367 | 0.498 | 0.399 | 0.335 | 0.422 | 0.369 | 0.258 | 0.353 | 0.314 |
| +*ImplicitSLIM* | 0.392 | 0.524 | 0.423 | 0.362 | 0.445 | 0.390 | 0.309 | 0.403 | 0.366 |
| WMF | 0.362 | 0.495 | 0.389 | 0.321 | 0.402 | 0.349 | | | |
| +*ImplicitSLIM* | 0.372 | 0.502 | 0.400 | 0.326 | 0.409 | 0.365 | | — | |
| **Matrix factorization with item embeddings regularization** | | | | | | | | | |
| CoFactor | 0.369 | 0.499 | 0.394 | 0.327 | 0.406 | 0.357 | | — | |
| GRALS | 0.376 | 0.505 | 0.401 | 0.335 | 0.416 | 0.365 | 0.201 | 0.275 | 0.245 |
| **Linear regression** | | | | | | | | | |
| SLIM | 0.370 | 0.495 | 0.401 | 0.347 | 0.428 | 0.379 | | — | |
| EASE | 0.391 | 0.521 | 0.420 | 0.362 | 0.445 | 0.393 | **0.333** | **0.428** | **0.389** |
| PLRec | 0.394 | 0.527 | 0.426 | 0.357 | 0.441 | 0.390 | 0.286 | 0.383 | 0.344 |
| +*ImplicitSLIM* | 0.391 | 0.522 | 0.423 | 0.358 | 0.440 | 0.390 | 0.310 | 0.406 | 0.364 |
| **Nonlinear autoencoders** | | | | | | | | | |
| CDAE | 0.391 | 0.523 | 0.418 | 0.343 | 0.428 | 0.376 | 0.188 | 0.283 | 0.237 |
| MultVAE | 0.395 | 0.537 | 0.426 | 0.351 | 0.444 | 0.386 | 0.266 | 0.364 | 0.316 |
| RaCT | 0.403 | 0.543 | 0.434 | 0.357 | 0.450 | 0.392 | 0.268 | 0.364 | 0.319 |
| RecVAE | 0.414 | 0.553 | 0.442 | 0.361 | 0.452 | 0.394 | 0.276 | 0.374 | 0.326 |
| +*ImplicitSLIM* | **0.419** | **0.559** | 0.447 | 0.365 | 0.455 | 0.398 | 0.291 | 0.391 | 0.342 |
| H+Vamp(Gated) | 0.413 | 0.551 | 0.445 | **0.377** | **0.463** | 0.407 | 0.289 | 0.381 | 0.342 |
| +*ImplicitSLIM* | 0.417 | 0.555 | **0.450** | **0.378** | **0.464** | **0.410** | 0.292 | 0.386 | 0.347 |

Table 2: Experimental results for GCN-based models.

| Model | Yelp2018 | | MovieLens-1M | |
|---|---|---|---|---|
| | Recall@20 | NDCG@20 | Recall@20 | NDCG@20 |
| LightGCN | 0.0649 | 0.0530 | 0.2576 | 0.2427 |
| UltraGCN | 0.0683 | 0.0561 | **0.2787** | 0.2642 |
| UltraGCN + *ImplicitSLIM* (users) | 0.0689 | 0.0568 | 0.2778 | 0.2648 |
| UltraGCN + *ImplicitSLIM* (items) | **0.0692** | **0.0573** | **0.2790** | **0.2659** |

Table 3: Experimental results for BERT4Rec.

| Model | # of epochs | MovieLens-1M | | | MovieLens-20M | | |
|---|---|---|---|---|---|---|---|
| | | Recall @5 | Recall @10 | NDCG @10 | Recall @5 | Recall @10 | NDCG @10 |
| BERT4Rec | 30 | 0.612 | 0.729 | 0.511 | 0.895 | 0.952 | 0.780 |
| BERT4Rec + *ImplicitSLIM* | | 0.660 | 0.765 | 0.549 | 0.904 | 0.959 | 0.789 |
| BERT4Rec | 200 | 0.645 | 0.758 | 0.545 | 0.901 | 0.953 | 0.790 |
| BERT4Rec + *ImplicitSLIM* | | **0.671** | **0.771** | **0.564** | **0.910** | **0.961** | **0.798** |

Table 1 also shows that *ImplicitSLIM* significantly improves performance (by the one-tailed test at the 95% confidence level) compared to all corresponding baselines except H+Vamp(Gated) on the *Netflix Prize* dataset in terms of *Recall@k*. Another valuable result is that deep models trained with ImplicitSLIM need about half as much time to achieve the best scores of deep models trained as is. By combining *ImplicitSLIM* with nonlinear variational autoencoders, RecVAE and H+Vamp(Gated), we have been able to improve over state of the art results on the *MovieLens-20M* and *Netflix Prize* datasets. Note that *ImplicitSLIM* together with RecVAE (or H+Vamp(Gated)) performs better on these datasets than both RecVAE and H+Vamp(Gated) and EASE. RecVAE with *ImplicitSLIM* performs on par with improved MF on the *Million Songs Dataset*, but with a much smaller embedding

dimension. Figure 2 shows sample convergence plots of UltraGCN and RecVAE with and without *ImplicitSLIM* for three select cases; periodic drops on the plots correspond to epochs when item embeddings were updated with *ImplicitSLIM*.

In our experiments, using *user* embeddings from *ImplicitSLIM* has not led to performance improvements, but this may be due to the strong generalization setting employed in most experiments.

Further, we have applied *ImplicitSLIM* to UltraGCN (Mao et al., 2021), a state-of-the-art GCN-based model, in the setup by He et al. (2020). Table 2 shows that we have succeeded in improving the performance by applying *ImplicitSLIM* to user embeddings, although *ImplicitSLIM* applied to item embeddings shows a significantly larger improvement. We note that both *ImplicitSLIM* and GCN-related models rely significantly on the diffusion of embeddings, so it is actually a bit surprising (albeit very encouraging) that *ImplicitSLIM* has been able to improve the performance of UltraGCN. Finally, we have applied *ImplicitSLIM* to BERT4Rec (Sun et al., 2019), a sequential model. Table 3 shows that although *ImplicitSLIM* does not take order into account, it can still be useful for sequential models. Namely, *ImplicitSLIM* allows BERT4Rec to achieve its best results in significantly fewer iterations, which is important for such heavy-weight models, and also improves its final performance. Appendix E shows our evaluation study in more details, including a detailed evaluation of specific models, runtime statistics, influence of *ImplicitSLIM* on unpopular items, convergence plots etc.

## 7 RELATED WORK

Matrix factorization (MF) (Mnih & Salakhutdinov, 2007; Bell et al., 2007; Koren & Bell, 2011) is a standard but still competitive CF baseline. Simple but high-performing alternative approaches include Sparse Linear Methods (SLIM) (see Section 2); they are difficult to scale so there exist low-rank variations such as factored item similarity models (FISM) (Kabbur et al., 2013). Other approaches reduce the problem to low-dimensional regression (Sedhain et al., 2016) (see Section 5) and perform low-dimensional decompositions of SLIM-like models (Jin et al., 2021). Nonlinear generalizations include autoencoder-based models that learn to map user feedback to user embeddings instead of learning the embedding matrix explicitly. Early approaches used shallow autoencoders (Sedhain et al., 2015; Wu et al., 2016), and recently variational autoencoders have led to better models (Liang et al., 2018; Lobel et al., 2019; Kim & Suh, 2019; Shenbin et al., 2020; Mirvakhabova et al., 2020). LRR (Jin et al., 2021) appears to be the model most similar to ours; it first computes the item-item similarity matrix and then performs its low-rank approximation, avoiding explicit computation of the similarity matrix by reducing the inversion of a large matrix to computing top eigenvectors; we reduce it to inverting a low-dimensional matrix. However, *ImplicitSLIM* has a different motivation than LRR and different usage. Graph convolutional networks (GCN) are relatively new in collaborative filtering; NFCF (Wang et al., 2019) and LightGCN (He et al., 2020) were computationally heavy, and recent approaches such as GF-CF (Shen et al., 2021) and UltraGCN (Mao et al., 2021) improved both performance and computational efficiency. Related approaches that regularize embeddings based on item-item interactions are discussed in Section 4. Dimensionality reduction methods can be either linear (e.g., PCA) or nonlinear, e.g., LLE (Roweis & Saul, 2000; Ghojogh et al., 2020) and ISOMAP (Tenenbaum et al., 2000)/ Modern nonlinear dimensionality reduction with VAEs (Kingma & Welling, 2014) has been successfully applied to collaborative filtering, but in a different way than the use of LLE in this work.

## 8 CONCLUSION

In this work, we have presented *ImplicitSLIM*, a novel approach based on the EASE and LLE models that can be used to initialize and/or regularize item and user embeddings for collaborative filtering models. We have shown that *ImplicitSLIM* can improve many existing models in terms of both performance and computational efficiency. We have applied *ImplicitSLIM* to MF, autoencoder-based, and graph-based models, and in most cases shown consistent improvements over the corresponding basic versions, achieving new state of the art results on classical datasets when applied to nonlinear autoencoder-based models. We propose *ImplicitSLIM* as a generic approach able to improve many collaborative filtering models.

REPRODUCIBILITY STATEMENT

All datasets used in our paper are publicly available, either together with scripts that split datasets into train/validation/test subsets (MovieLens-20M, Netflix Prize Dataset, and Million Songs Dataset) or already available divided into train/test subsets (Yelp2018 and Movielens-1M). Source code for the implementation of *ImplicitSLIM* sufficient to reproduce the most important results of this paper is submitted as supplementary materials. The source code is available on *GitHub*: `https://github.com/ilya-shenbin/ImplicitSLIM`. Furthermore, pseudocode of downstream methods is presented in Appendix D. The hyperparameter search process is described in Appendix C, and all necessary derivations are presented in Appendix A.

ACKNOWLEDGMENTS

This work was performed at the Saint Petersburg Leonhard Euler International Mathematical Institute and supported by the Ministry of Science and Higher Education of the Russian Federation (Agreement no. 075-15-2022-289, dated 06/04/2022).

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

## A    PROOFS AND DERIVATIONS

### A.1    DEFINITION OF LLE

In this section we introduce LLE as it was done in the original works by Roweis & Saul (2000); Saul & Roweis (2003). We denote by $\mathbf{D}_{i*}$ the $i$-th data sample, by $\mathbf{W}_{i*}$ its local coordinate vector, and by $\mathrm{NN}(i)$ the set of indices for nearest neighbors of the $i$-th data sample; $n$ is the number of data samples. The first step is to find the matrix $\hat{\mathbf{W}}$:

$$\hat{\mathbf{W}} = \mathrm{argmin}_{\mathbf{W}} \ \sum_i \left\| \mathbf{D}_{i*} - \sum_{j \in \mathrm{NN}(i)} \mathbf{W}_{ij} \mathbf{D}_{j*} \right\|_2^2 \ \text{s.t.} \ \sum_j \mathbf{W}_{ij} = 1, \tag{17}$$

where $\hat{\mathbf{W}}_{ij} = 0$ if $j \notin \mathrm{NN}(i)$. On the second step, LLE finds the embedding matrix $\hat{\mathbf{Y}}$ given the sparse matrix $\hat{\mathbf{W}}$:

$$\hat{\mathbf{Y}} = \mathrm{argmin}_{\mathbf{Y}} \ \sum_i \left\| \mathbf{Y}_{i*} - \sum_j \hat{\mathbf{W}}_{ij} \mathbf{Y}_{j*} \right\|_2^2$$
$$\text{s.t.} \ \frac{1}{n} \sum_i \mathbf{Y}_{i*} \mathbf{Y}_{i*}^\top = \mathbf{I}, \quad \sum_i \mathbf{Y}_{ij} = 0. \tag{18}$$

Denoting $\mathbf{D} = \mathbf{X}^\top$, $\mathbf{W} = \mathbf{B}^\top$, and $\mathbf{Y} = \mathbf{V}^\top$ we can rewrite (17) and (18) in this notation as

$$\hat{\mathbf{B}} = \mathrm{argmin}_{\mathbf{B}} \ \sum_i \left\| \mathbf{X}_{*i} - \sum_{j \in \mathrm{NN}(i)} \mathbf{X}_{*j} \mathbf{B}_{ji} \right\|_2^2 \ \text{s.t.} \ \sum_j \mathbf{B}_{ji} = 1, \tag{19}$$

$$\hat{\mathbf{V}} = \mathrm{argmin}_{\mathbf{V}} \ \sum_i \left\| \mathbf{V}_{*i} - \sum_j \mathbf{V}_{*j} \hat{\mathbf{B}}_{ji} \right\|_2^2$$
$$\text{s.t.} \ \frac{1}{n} \sum_i \mathbf{V}_{*i} \mathbf{V}_{*i}^\top = \mathbf{I}, \quad \sum_i \mathbf{V}_{ji} = 0. \tag{20}$$

Rewriting formulas (19) and (20) in matrix form whenever possible, we derive (3) and (4) respectively.

### A.2    DERIVATION OF SLIM-LLE

To get the item-item interaction matrix from the first step

$$\hat{\mathbf{B}} = \mathrm{argmin}_{\mathbf{B}} \ \|\mathbf{X} - \mathbf{X}\mathbf{B}\|_F^2 + \lambda\|\mathbf{B}\|_F^2 \quad \text{s.t.} \ \mathbf{B}^\top \mathbf{1} = \mathbf{1}, \ \mathrm{diag}\,\mathbf{B} = \mathbf{0}, \tag{21}$$

we have to solve the following system of linear equations:

$$\begin{cases} \dfrac{\partial}{\partial \mathbf{B}} \left( \|\mathbf{X} - \mathbf{X}\mathbf{B}\|_F^2 + \lambda\|\mathbf{B}\|_F^2 + 2\gamma^T \mathrm{diag}\,(\mathbf{B}) + 2\kappa^T (\mathbf{B}^T \mathbf{1} - \mathbf{1}) \right) = 0, \\ \mathbf{B}^\top \mathbf{1} = \mathbf{1}, \\ \mathrm{diag}\,\mathbf{B} = \mathbf{0}. \end{cases} \tag{22}$$

The solution is

$$\hat{\mathbf{B}} = \mathbf{I} - \left( \hat{\mathbf{P}} - \frac{\hat{\mathbf{P}}\mathbf{1}(\hat{\mathbf{P}}\mathbf{1})^T}{\mathbf{1}^T \hat{\mathbf{P}}\mathbf{1}} \right) \mathrm{diagMat} \left( \mathbf{1} \oslash \mathrm{diag} \left( \hat{\mathbf{P}} - \frac{\hat{\mathbf{P}}\mathbf{1}(\hat{\mathbf{P}}\mathbf{1})^T}{\mathbf{1}^T \hat{\mathbf{P}}\mathbf{1}} \right) \right),$$

where $\hat{\mathbf{P}} \stackrel{\mathrm{def}}{=} (\mathbf{X}^\top \mathbf{X} + \lambda \mathbf{I})^{-1}$.

### A.3   DIAGONAL OF THE INVERSE OF A MATRIX

We begin by representing the inverse of a matrix $\mathbf{A}$ as an infinite sum derived from the Neumann series:

$$\mathbf{A}^{-1} = \sum_{n=0}^{\infty} (\mathbf{I} - \mathbf{A})^n. \tag{23}$$

We note that $\mathbf{B}^{-1} = \mathbf{D}^{-1}\mathbf{D}\mathbf{B}^{-1} = \mathbf{D}^{-1}\left(\mathbf{B}\mathbf{D}^{-1}\right)^{-1}$, where $\mathbf{D} \overset{\text{def}}{=} \mathrm{diagMat}(\mathrm{diag}\,\mathbf{B})$, and substitute $\mathbf{A} = \mathbf{B}\mathbf{D}^{-1}$ into (23) to obtain

$$\mathbf{B}^{-1} = \mathbf{D}^{-1} \sum_{n=0}^{\infty} \left(\mathbf{I} - \mathbf{B}\mathbf{D}^{-1}\right)^n. \tag{24}$$

We now approximate $\mathbf{B}^{-1}$ with the first two terms of the Neumann series:

$$\mathbf{B}^{-1} \approx \mathbf{D}^{-1} \left(\mathbf{I} + \left(\mathbf{I} - \mathbf{B}\mathbf{D}^{-1}\right)\right).$$

Since we need only $\mathrm{diag}\,\mathbf{B}^{-1}$ rather than the full matrix $\mathbf{B}^{-1}$, we simplify the result to

$$\mathrm{diag}\,\mathbf{B}^{-1} \approx \mathrm{diag}(\mathbf{D}^{-1}\left(\mathbf{I} + \left(\mathbf{I} - \mathbf{B}\mathbf{D}^{-1}\right)\right)) = 2\mathbf{D}^{-1} - \mathrm{diag}(\mathbf{D}^{-1}\mathbf{B}\mathbf{D}^{-1}) = \mathbf{D}^{-1}.$$

Next we show that the series (24) converges:

$$\left\| \sum_{n=0}^{\infty} \left(\mathbf{I} - \mathbf{B}\mathbf{D}^{-1}\right)^n \right\|_1 \leq \sum_{n=0}^{\infty} \left\| \mathbf{I} - \mathbf{B}\mathbf{D}^{-1} \right\|_1^n.$$

As we defined earlier, $\mathbf{B}_{ij} = \mathbf{X}_{*i}^\top \mathbf{X}_{*j} + \lambda \llbracket i = j \rrbracket$, $\mathbf{X} \in \{0,1\}^{U \times I}$, $\lambda > 0$, and $\mathbf{D}_{ij} = \left(\mathbf{X}_{*i}^\top \mathbf{X}_{*j} + \lambda\right) \llbracket i = j \rrbracket$, where $\llbracket \cdot \rrbracket$ is the indicator function. Then

$$\left\| \mathbf{I} - \mathbf{B}\mathbf{D}^{-1} \right\|_1 = \max_j \sum_i \left| \llbracket i = j \rrbracket - \left(\mathbf{B}\mathbf{D}^{-1}\right)_{ij} \right| =$$

$$= \max_j \sum_i \left| \llbracket i = j \rrbracket - \frac{\mathbf{X}_{*i}^\top \mathbf{X}_{*j} + \lambda \llbracket i = j \rrbracket}{\mathbf{X}_{*j}^\top \mathbf{X}_{*j} + \lambda} \right| =$$

$$= \max_j \sum_i \left| \frac{\left(\llbracket i = j \rrbracket \mathbf{X}_{*j}^\top - \mathbf{X}_{*i}^\top\right) \mathbf{X}_{*j}}{\mathbf{X}_{*j}^\top \mathbf{X}_{*j} + \lambda} \right|$$

$$= \max_j \frac{\sum_{i:i \neq j} \mathbf{X}_{*i}^\top \mathbf{X}_{*j}}{\mathbf{X}_{*j}^\top \mathbf{X}_{*j} + \lambda} \leq \max_j \frac{(I-1)\mathbf{X}_{*j}^\top \mathbf{X}_{*j}}{\mathbf{X}_{*j}^\top \mathbf{X}_{*j} + \lambda}.$$

Thus, if $\lambda > (I-2)\max_j \left(\mathbf{X}_{*j}^\top \mathbf{X}_{*j}\right)$ then $\left\| \mathbf{I} - \mathbf{B}\mathbf{D}^{-1} \right\|_1 < 1$, and the series in (24) converges.

### A.4   LLE AND GRAPH REGULARIZATION

We can represent the *SLIM regularizer* (15) as a graph regularizer (5):

$$\|\mathbf{Q} - \mathbf{Q}\hat{\mathbf{B}}\|_F^2 = \|\mathbf{Q}(\mathbf{I} - \hat{\mathbf{B}})\|_F^2 = \mathrm{tr}\left(\mathbf{Q}(\mathbf{I} - \hat{\mathbf{B}})(\mathbf{I} - \hat{\mathbf{B}})^T \mathbf{Q}^T\right) \overset{\text{def}}{=} \mathrm{tr}\left(\mathbf{Q}\hat{\mathbf{L}}\mathbf{Q}^T\right),$$

Here $\hat{\mathbf{L}}$ is a positive semi-definite matrix since it is a Gram matrix. However, in general $\hat{\mathbf{L}}$ is not a graph Laplacian matrix.

Assuming that $\hat{\mathbf{B}}^\top \mathbf{1} = \mathbf{1}$ and $\mathrm{diag}\,\hat{\mathbf{B}} = \mathbf{0}$, we get

$$\hat{\mathbf{L}}\mathbf{1} = (\mathbf{I} - \hat{\mathbf{B}})(\mathbf{I} - \hat{\mathbf{B}})^T \mathbf{1} = (\mathbf{I} - \hat{\mathbf{B}})\mathbf{0} = \mathbf{0}.$$

Since $\hat{\mathbf{L}}$ is also a symmetric matrix, the sum of any row or column is equal to zero. As a result, there exists such an adjacency matrix $\hat{\mathbf{A}}$ that $\hat{\mathbf{L}}$ is the corresponding graph Laplacian matrix, and

$$\hat{\mathbf{A}} = \mathrm{diagMat}(\mathrm{diag}(\hat{\mathbf{L}})) - \hat{\mathbf{L}}.$$

---

**Algorithm 1:** Pseudocode for **vanilla** matrix factorization

---

**Data:** dataset $\mathbf{X}$, number of iterations $k$, hyperparameters $r_p, r_q$
**Result:** item embeddings matrix $\mathbf{Q}$
$\mathbf{Q} \leftarrow$ standart Gaussian noise ;
**for** $i \leftarrow 1$ **to** $k$ **do**
   | $\mathbf{P} \leftarrow \mathrm{argmin}_{\mathbf{P}} \ \mathcal{L}_{\mathrm{MF}}(\mathbf{X}_{\mathrm{train}}, \mathbf{Q}, r_p)$;
   | $\mathbf{Q} \leftarrow \mathrm{argmin}_{\mathbf{Q}} \ \mathcal{L}_{\mathrm{MF}}(\mathbf{X}_{\mathrm{train}}, \mathbf{P}, r_q)$;
   | *evaluate(*$\mathbf{X}_{valid}, \mathbf{Q}, r_p$*)*;
   | **if** *current validation score < the best validation score* **then**
      | break;
   | **end**
**end**

---

Note that some values in $\hat{\mathbf{A}}$ could be negative; while it is uncommon to consider a graph Laplacian matrix for a graph with negative weight edges, such matrices have been introduced in some papers, in particular by Zelazo & Bürger (2014). Fortunately, even despite the presence of negative values in the adjacency matrix the graph regularizer is still equal to the weighted sum of pairwise distances (5).

As a result, we claim that the *SLIM regularizer* with matrix $\hat{\mathbf{B}}$ taken from SLIM-LLE is equal to the graph regularizer. Unfortunately, this equality does not hold for the matrix $\hat{\mathbf{B}}$ taken from EASE.

# B  ADDITIONAL DETAILS ON BASELINES

## B.1  MATRIX FACTORIZATION

The MF model we used in our experiments is defined by the following loss function:

$$\mathcal{L}_{\mathrm{MF}}(\mathbf{P}, \mathbf{Q}) = \|\mathbf{X} - \mathbf{P}^\top \mathbf{Q}\|_F^2 + r_p \|\mathbf{P}\|_F^2 + r_q \|\mathbf{Q}\|_F^2, \tag{25}$$

where $\mathbf{P} \in \mathbb{R}^{L \times U}$ is the user embeddings matrix, $\mathbf{Q} \in \mathbb{R}^{L \times I}$ is the item embeddings matrix, and $r_p$ and $r_q$ are regularization coefficients. $\mathcal{L}_{\mathrm{MF}}(\mathbf{P}, \mathbf{Q})$ can be optimized with alternating least squares (ALS).

We used eALS[5] as an implementation of WMF, but had to adapt it to the strong generalization setting, so that the model would be comparable to others.

For MF and WMF, we can easily obtain embeddings of held-out users from validation/test sets in the same way we update user embeddings during training using the closed form solution for user embeddings given fixed item embeddings and held-out validation/test feedback matrix.

## B.2  MATRIX FACTORIZATION WITH ITEM EMBEDDINGS REGULARIZATION

CoFactor was evaluated in a different setup in the original papers, so we had to re-evaluate it. We use a publicly available implementation of CoFactor[6], but similarly to eALS we have had to adapt it to the strong generalization setting.

Since there is no publicly available implementation of GRALS, we have implemented it independently. We use the cooccurrence matrix $\mathbf{X}^\top \mathbf{X}$ as the adjacency matrix with diagonal elements zeroed. Following the experimental results of He et al. (2020), we use the symmetrically normalized graph Laplacian. Graph regularization was applied to item embeddings only to be consistent with other experiments.

In order to update item embeddings in GRALS we have to solve the Sylvester equation, which requires us to compute the eigenvectors of a large matrix on every step of the ALS algorithm. To avoid this computation, we propose to update item embeddings using gradient descent with optimal step size search, which just slightly hurts performance in terms of ranking metrics while being much more computationally efficient.

---

[5]`https://github.com/newspicks/eals`
[6]`https://github.com/dawenl/cofactor`

---

**Algorithm 2:** Pseudocode for the **vanilla** PLRec

---

**Data:** dataset $\mathbf{X}$, hyperparameter $r_q$
**Result:** item embeddings matrix $\mathbf{Q}$
$\mathbf{W} \leftarrow$ initialize using SVD ;
$\mathbf{Q} \leftarrow \text{argmin}_{\mathbf{Q}} \ \mathcal{L}_{\text{PLREC}}(\mathbf{X}_{\texttt{train}}, \mathbf{W}, r_q)$;
*evaluate*$(\mathbf{X}_{valid}, \mathbf{Q}, r_p)$;

---

### B.3 LINEAR MODELS

We use SLIM (Ning & Karypis, 2011) and EASE (Steck, 2019) as baselines, taking the scores of SLIM from (Liang et al., 2018), and scores of EASE from (Steck, 2019).

For our experiments we have reimplemented PLRec, we consider the following objective function:

$$\mathcal{L}_{\text{PLREC}}(\mathbf{Q}) = \|\mathbf{X} - \mathbf{X}\mathbf{W}^\top \mathbf{Q}\|_F^2 + r_q\|\mathbf{Q}\|_F^2. \tag{26}$$

There are two item embedding matrices in this model: $\mathbf{W}$ projects user ratings into a low-dimensional space and $\mathbf{Q}$ maps user embeddings to predicted user ratings. Matrix $\mathbf{W}$ is assumed to be initialized by the SVD item embeddings matrix, and is fixed during training.

We have found that normalizing the columns of $\mathbf{X}$ before projecting into a low-dimensional space significantly improves the performance. Specifically, in our experiments we replaced $\mathbf{W}^\top$ in (26) with $\mathbf{N}^{-1}\mathbf{W}^\top$, where $\mathbf{N}$ is a diagonal matrix with $\mathbf{N}_{ii} = \|\mathbf{X}_{*i}\|_1^n$, where $n$ is a parameter to be tuned during cross-validation.

### B.4 NONLINEAR AUTOENCODERS

We consider several nonlinear autoencoders as baselines: CDAE (Wu et al., 2016), MultVAE (Liang et al., 2018), RaCT (Lobel et al., 2019), RecVAE (Shenbin et al., 2020), and H+Vamp(Gated) (Kim & Suh, 2019). Scores for these models were taken from the corresponding papers, except for CDAE where we took them from (Liang et al., 2018). Also, scores of H+Vamp(Gated) on Million Song Dataset were not published in original paper, so we get the scores ourselves using original implementation of this model.

## C HYPERPARAMETER SEARCH

We use the *BayesianOptimization* library (Nogueira, 2014) for hyperparameter search. At every step of Bayesian optimization, we evaluate the model with the current set of hyperparameters. When the hyperparameter search procedure ends, we select the set of hyperparameters that maximizes the validation score; then we use the selected set of hyperparameters to evaluate the model on test data.

Learning a large number of models is time-consuming for models with high-dimensional embeddings. In order to speed it up, we start hyperparameter search with low-dimensional embeddings. When we terminate the search procedure for the current dimensionality, we increase the dimensionality and begin a new hyperparameter search, but begin to evaluate the model on the set of hyperparameters that was chosen as best at the previous dimensionality. This narrows down the search in larger dimensions and can find sufficiently good hyperparameters much faster, especially for heavyweight models.

We perform at least 200 steps of Bayesian optimization for every model, with the exception of most heavyweight models, for which we introduce an additional external restriction of a ten hour limit per model.

The heaviest setup for training MF and PLRec is **SLIM-LLE init**. Unlike the others, it requires huge matrix inversions and computation of eigenvectors for the smallest eigenvalues. As a result, computational costs for this setting are much higher, especially when the dimensionality of embeddings is large. Hence we have limited the evaluation of this setup on the Million Song Dataset with embedding dimensions equal to 1024 and lower. Additionally, we have precomputed LLE-SLIM

---

**Algorithm 3:** Pseudocode for **ImplicitSLIM init+reg** matrix factorization

---

**Data:** dataset $\mathbf{X}$, number of iterations $k$, hyperparameters $r_p$, $r_q$, $s_q$, $\lambda$, $\alpha$
**Result:** item embeddings matrix $\mathbf{Q}$
$\mathbf{Q} \leftarrow$ standart Gaussian noise ;
**for** $i \leftarrow 1$ **to** $k$ **do**
    $\mathbf{V} \leftarrow$ *ImplicitSLIM($\mathbf{X}_{train}$, $\mathbf{Q}$, $\lambda$, $\alpha$)*;
    **if** $i = 1$ **then**
        $\mathbf{Q} \leftarrow \mathbf{V}$;
    **end**
    $\mathbf{P} \leftarrow \mathrm{argmin}_{\mathbf{P}} \ \mathcal{L}_{\text{MF-ImplicitSLIM}}(\mathbf{X}_{\text{train}}, \mathbf{Q}, r_p)$;
    $\mathbf{Q} \leftarrow \mathrm{argmin}_{\mathbf{Q}} \ \mathcal{L}_{\text{MF-ImplicitSLIM}}(\mathbf{X}_{\text{train}}, \mathbf{P}, \mathbf{V}, r_q, s_q)$;
    *evaluate($\mathbf{X}_{valid}$, $\mathbf{Q}$, $r_p$)*;
    **if** *current validation score < the best validation score* **then**
        break;
    **end**
**end**

---

**Algorithm 4:** Pseudocode for **ImplicitSLIM init** matrix factorization

---

**Data:** dataset $\mathbf{X}$, number of iterations $k$, hyperparameters $r_p$, $\lambda$, $\alpha$
**Result:** item embeddings matrix $\mathbf{Q}$
$\mathbf{Q} \leftarrow$ standart Gaussian noise ;
**for** $i \leftarrow 1$ **to** $k$ **do**
    $\mathbf{Q} \leftarrow$ *ImplicitSLIM($\mathbf{X}_{train}$, $\mathbf{Q}$, $\lambda$, $\alpha$)*;
    *evaluate($\mathbf{X}_{valid}$, $\mathbf{Q}$, $r_p$)*;
    **if** *current validation score < the best validation score* **then**
        break;
    **end**
**end**

---

**Algorithm 5:** Pseudocode for **SLIM-LLE init** matrix factorization

---

**Data:** dataset $\mathbf{X}$, number of iterations $k$, hyperparameters $r_p$, $\lambda$, $\alpha$
**Result:** item embeddings matrix $\mathbf{Q}$
$\mathbf{Q} \leftarrow$ *SLIM-LLE($\mathbf{X}_{train}$, $\lambda$)*;
*evaluate($\mathbf{X}_{valid}$, $\mathbf{Q}$, $r_p$)*;

---

embeddings for various values of $\lambda$ on a logarithmic grid with step $0.1$, and we have not computed embeddings for other values of $\lambda$ during hyperparameter tuning.

## D  ADDITIONAL DETAILS ON IMPLICITSLIM WITH OTHER MODELS

### D.1  IMPLICITSLIM WITH MATRIX FACTORIZATION AND PLREC

We have evaluated *ImplicitSLIM* in application to matrix factorization and PLRec. To describe these setups in detail, we present the pseudocode for all of them. We define the loss function for **ImplicitSLIM init+reg** as follows:

$$\mathcal{L}_{\text{MF-ImplicitSLIM}}(\mathbf{P}, \mathbf{Q}, \mathbf{V}) = \|\mathbf{X} - \mathbf{P}^\top \mathbf{Q}\|_F^2 + s_q \|\mathbf{V} - \mathbf{Q}\|_F^2 + r_p \|\mathbf{P}\|_F^2 + r_q \|\mathbf{Q}\|_F^2 \quad (27)$$

Regularizer $s_q \|\mathbf{V} - \mathbf{Q}\|_F^2$ is used instead of $s_q \|(\mathbf{V} - \mathbf{Q})\mathbf{Q}^\top\|_F^2$ to get a closed form solution.

In order to present Algorithm 6, we first define the $\mathcal{L}_{\text{PLREC-ImplicitSLIM}}$ loss as

$$\mathcal{L}_{\text{PLREC-ImplicitSLIM}}(\mathbf{Q}, \mathbf{V}) = \|\mathbf{X} - \mathbf{X}\mathbf{W}^\top \mathbf{Q}\|_F^2 + r_q \|\mathbf{Q}\|_F^2 + s_q \|\mathbf{V} - \mathbf{Q}\|_F^2.$$

---

**Algorithm 6:** Pseudocode for **ImplicitSLIM init+reg** PLRec

---

**Data:** dataset $\mathbf{X}$, number of iterations $k$, hyperparameters $r_q$, $s_q$, $\lambda$, $\alpha$
**Result:** item embeddings matrix $\mathbf{Q}$
$\mathbf{V} \leftarrow$ standart Gaussian noise ;
**for** $i \leftarrow 1$ **to** $k$ **do**
    $\mathbf{V} \leftarrow$ *ImplicitSLIM($\mathbf{X}_{train}$, $\mathbf{V}$, $\lambda$, $\alpha$)*;
    $\mathbf{V} \leftarrow$ *orthogonalize($\mathbf{V}$)*;
    $\mathbf{W} \leftarrow \mathbf{V}$;
    $\mathbf{Q} \leftarrow \text{argmin}_{\mathbf{Q}} \; \mathcal{L}_{\text{PLREC-IMPLICITSLIM}}(\mathbf{X}_{\text{train}}, \mathbf{W}, \mathbf{V}, r_q, s_q)$;
    *evaluate($\mathbf{X}_{valid}$, $\mathbf{Q}$, $r_p$)*;
    **if** *current validation score < the best validation score* **then**
        break;
    **end**
**end**

---

**Algorithm 7:** Pseudocode for **SLIM-LLE init** PLRec

---

**Data:** dataset $\mathbf{X}$, hyperparameters $r_q$, $\lambda$
**Result:** item embeddings matrix $\mathbf{Q}$
$\mathbf{W} \leftarrow$ *SLIM-LLE($\mathbf{X}_{train}$, $\lambda$)*;
$\mathbf{Q} \leftarrow \text{argmin}_{\mathbf{Q}} \; \mathcal{L}_{\text{PLREC}}(\mathbf{X}_{\text{train}}, \mathbf{W}, r_q)$;
*evaluate($\mathbf{X}_{valid}$, $\mathbf{Q}$, $r_p$)*;

---

Algorithms in this section return only item embedding matrices since we are working in the *strong generalization setting*; the user embedding matrix is not required for further evaluations.

In the main text, we approximated the user-item interactions matrix $\mathbf{X}$ with a low-rank matrix $\mathbf{P}^\top \mathbf{Q}$. In our experiments we also used the matrix $\mathbf{P}^\top \mathbf{Q} + \mathbf{1}\mathbf{b}^\top$ instead, i.e., employed a bias vector. Using a bias vector has led to better results in some experiments. For both matrix factorization and PLRec, we estimate $\mathbf{b}$ using maximum likelihood, and in both cases we set $\mathbf{b}_i$ as the mean value of elements of vector $\mathbf{X}_{*i}$.

In Section 3.2 it was proposed to set $\mathbf{A}$ equal to $\mathbf{Q}$. Since embeddings of unpopular items could be noisy, we propose to first set $\mathbf{A}$ equal to $\mathbf{Q}$ but then zero out columns in $\mathbf{A}$ that correspond to unpopular items; a threshold of item popularity here becomes another hyperparameter.

### D.2 ImplicitSLIM with VAE, GCN and BERT4Rec

The most successful of these models, namely RecVAE and H+Vamp(Gated), have also been evaluated with additional regularization via *ImplicitSLIM*. For this purpose, we used the original implementations of RecVAE[7] and H+Vamp(Gated)[8] and integrated *ImplicitSLIM* into them as follows: the item embeddings matrices from both encoder and decoder are updated with *ImplicitSLIM* once every several epochs (every 10 epochs for RecVAE, every 15 epochs for H+Vamp(Gated)); additionally, the dimension of embeddings has been increased to 400 for RecVAE; the rest of the setup is taken from original implementations.

According to Zhu et al. (2022), GF-CF (Shen et al., 2021) and UltraGCN (Mao et al., 2021) are state-of-the-art GCN-influenced models. However, GF-CF has no trainable embeddings, so we apply *ImplicitSLIM* to UltraGCN only. Unlike models we have mentioned above, UltraGCN is based on a different setup. It has trainable embeddings for both users and items, and the same users are considered in both train and test time. That allows us to apply *ImplicitSLIM* not only for item embeddings but also for user embeddings (using its official implementation[9]). In both cases we update embeddings once in several epochs, similarly to nonlinear autoencoders.

---

[7]https://github.com/ilya-shenbin/RecVAE
[8]https://github.com/psywaves/EVCF
[9]https://github.com/xue-pai/UltraGCN

Table 4: 1NN scores of embeddings obtained with different methods.

| Embedding dimensionality | 10 | 100 |
|---|---|---|
| Standard Gaussian noise | 0.423 | 0.423 |
| SVD | 0.634 | 0.695 |
| SLIM-LLE | 0.661 | 0.713 |
| *ImplicitSLIM* | 0.630 | 0.704 |
| MF | 0.620 | 0.693 |
| MF + *ImplicitSLIM* | 0.636 | 0.692 |
| RecVAE | 0.667 | 0.760 |
| RecVAE + *ImplicitSLIM* | 0.688 | 0.758 |

In order to show that *ImplicitSLIM* can be useful not only for collaborative filtering models in their classical form, we have also applied it to sequence-based recommendations. We have chosen BERT4Rec (Shen et al., 2021) as one of the most popular sequential model with strong performance. To perform our experiments we took its unofficial implementation[10], since the results shown by Shen et al. (2021) fail to reproduce with the official implementation, according to experiments performed by Petrov & Macdonald (2022). Item embedding matrices are updating during training every two epochs for the *MovieLens-1M* dataset and every 20 epochs for the *MovieLens-20M*, similarly to nonlinear autoencoders.

According to our experiments, *ImplicitSLIM* regularization (namely minimizing the distance between the current embedding matrix and the one updated by *ImplicitSLIM*) does not improve the performance of VAEs and GCNs compared to the approach proposed above. According to our intuition, it happens because embeddings updated by ImplicitSLIM become out of date after several stochastic gradient decent updates of current embedding vectors. Moreover, this form of *ImplicitSLIM* application is more computationally expensive.

### D.3 EVALUATION OF EMBEDDINGS WITH 1NN

In additional to our evaluation of *ImplicitSLIM* on the collaborative filtering downstream task, we have also evaluated extracted embeddings employing contextual information. Items from the *MovieLens-20M* dataset have genre labels, where some items could have several labels and others might have none. Items without labels were excluded for this experiment.

We evaluate the embeddings using a popular way to evaluate visualization methods based on the assumption that similar items should have similar labels. We define it as follows:

$$\frac{1}{I} \sum_{i=1}^{I} [\![\mathbf{Q}_{*i} \text{ and its nearest neighbor have at least one common label}]\!],$$

where $[\![\cdot]\!]$ is the indicator function and $\mathbf{Q}_{*i}$ is the embedding vector of the $i$th item. Results are presented in Table 4. For the sake of a fair comparison, labels were not used either when learning the embeddings or in tuning the hyperparameters. We use the same hyperparameters as for experiments shown in Table 1.

## E ADDITIONAL DETAILS ON RESULTS

### E.1 RUNTIME

Table 5 shows a runtime comparison of *ImplicitSLIM* and its variation with explicit computation of the matrix $\hat{\mathbf{B}}$ (that includes Step 1 and Step 2). The runtime of *ImplicitSLIM* significantly depends on the dimensionality of the embedding matrix, thus we have measured it for different dimensions. As we can see, using *ImplicitSLIM* instead of straightforward calculations drastically reduces computational time.

---

[10]`https://github.com/jaywonchung/BERT4Rec-VAE-Pytorch`

Table 5: Runtime of embedding extraction procedures, seconds.

| | Step 1 (1) | Step 2 (8) | *ImplicitSLIM* | | | | | | |
| | | | 64 | 128 | 256 | 512 | 1024 | 2048 | 4096 |
|---|---|---|---|---|---|---|---|---|---|
| ML-20M | 48.7 | 53.3 | 0.37 | 0.7 | 1.4 | 2.6 | 5.1 | 12.0 | 31.1 |
| Netflix | 67.4 | 78.1 | 1.7 | 3.4 | 6.6 | 12.1 | 23.0 | 52.0 | 133.2 |
| MSD | 389.9 | 452.9 | 1.4 | 2.5 | 7.7 | 17.5 | 32.7 | 64.5 | 132.0 |

Table 6: Evaluation of high, mid, and low popularity items, NDCG@100 scores

| Dataset | MovieLens-20M | | | Netflix Prize Dataset | | |
| Items poplarity | high | medium | low | high | medium | low |
|---|---|---|---|---|---|---|
| MF | 0.392 | 0.097 | 0.036 | 0.366 | 0.092 | 0.057 |
| MF + *ImplicitSLIM* | 0.416 | 0.110 | 0.040 | 0.383 | 0.138 | 0.089 |
| RecVAE | 0.444 | 0.192 | 0.092 | 0.402 | 0.209 | 0.157 |
| RecVAE + *ImplicitSLIM* | 0.449 | 0.180 | 0.086 | 0.405 | 0.213 | 0.165 |

Fig. 3 shows convergence plots for matrix factorization (MF), MF with *ImplicitSLIM* init (we call it MF + *ImplicitSLIM* here for short), and EASE. Since EASE is not an iterative model, its plot is presented as a single point. We also note that MF + *ImplicitSLIM* reaches its best scores in one or two iterations in these experiments. We show individual plots for MF and MF + *ImplicitSLIM* with different embedding dimensionality. As a result, Figure 3 shows that MF + *ImplicitSLIM* obtains better results than MF in terms of both metric scores and wall clock time. Moreover, MF + *ImplicitSLIM* is also competitive with EASE: while it obtains slightly lower scores in the quality metric, it contains much fewer parameters and needs less time to train in most cases.

### E.2 EVALUATION OF APPROXIMATIONS

To make *ImplicitSLIM* efficient in both computation time and memory, we have made several approximations and simplifications:

(a) the constraint in (6) is dropped;

(b) diagonal of the inverted matrix is approximated with (12);

(c) $\|(\mathbf{V} - \mathbf{Q})\mathbf{Q}^\top\|_F^2$ is used in (7) instead of $\|\mathbf{V} - \mathbf{Q}\|_F^2$.

We have compared the performance of *ImplicitSLIM* and its explicit version where approximations (a-c) are not used, using MF and RecVAE as base models on the *MovieLens-20M* and *Netflix* datasets. In all cases the difference in results was negligible, so they are not shown in Table 1, which validates the efficiency of approximations in *ImplicitSLIM*. We note, however, that the almost identical optimal scores were achieved at different hyperparameter values.

### E.3 EVALUATION ON UNPOPULAR ITEMS

We divided the items into groups of the same size: very popular, with medium popularity, and with low popularity, and then evaluated several methods on each of these groups. According to the results shown in Table 6, *ImplicitSLIM* generally boosts the performance for items with low and medium popularity more significantly than for very popular items. There is one exception, however: RecVAE evaluated on MovieLens-20M. In our opinion, this is an effect of zeroing out columns in matrix $\mathbf{A}$ that correspond to unpopular items, as we describe it in Appendix D.1, which hurts the performance for unpopular items in this case.

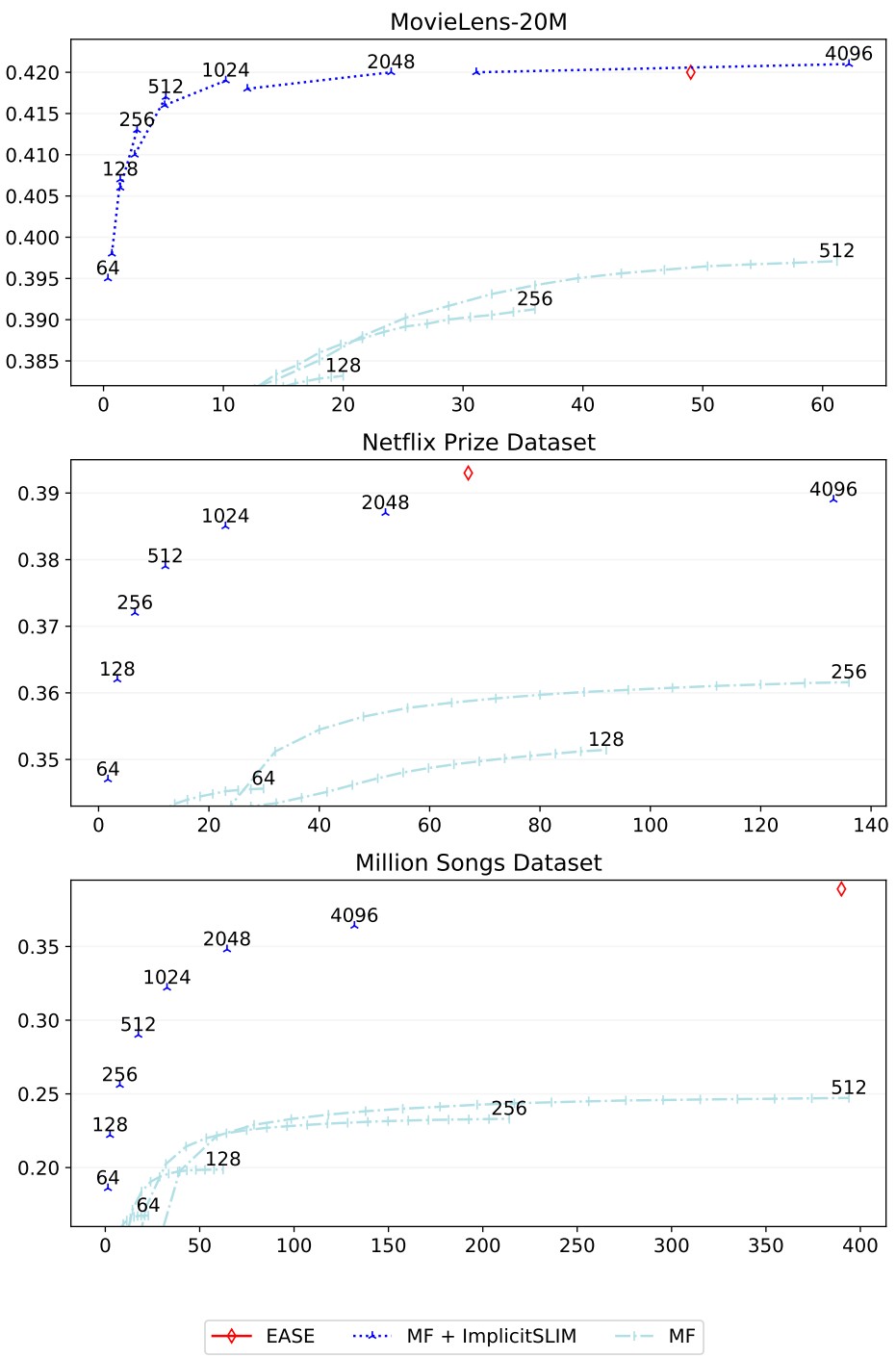

Figure 3: Convergence plots for MF, MF + *ImplicitSLIM*, and EASE; the X-axis shows wall-clock time is seconds, the Y-axis shows NDCG@100; MF and MF + *ImplicitSLIM* have been evaluated with different embedding dimensions shown in the figure.

