# OpenReview forum: "ImplicitSLIM and How it Improves Embedding-based Collaborative Filtering"
_ICLR.cc/2024/Conference — ICLR 2024 poster_

### Official Review · Reviewer_dkLJ · 2023-10-18

**Soundness:** 4 excellent
**Presentation:** 2 fair
**Contribution:** 3 good
**Rating:** 6
**Confidence:** 4

**Summary:**

This paper focuses on an influencial autoencoder-based collaborative filtering model SLIM and draws inspiration from its optimization objective and correspondingly proposes ImplicitSLIM. ImplicitSLIM combines the advantages of SLIM's objective function and Locally Linear Embeddings. ImplicitSLIM also introduces a novel regularization item and an initialization method for embedding-based collaborative filtering models. Experimental results are presented to demonstrate the effectiveness of the proposed model.

**Strengths:**

+ The authors propose a novel and general method that enhances performance of embedding-based CF models, which has practical significance.
+ The theoretical analysis of existing works on autoencoder-based models is persuasive and easy to follow. The paper provides a clear and insightful review of existing methods from an optimization perspective.
+ ImplicitSLIM is well-motivated and presents a novel solution for embedding-based CF.
+ The paper is generally well-written, ensuring readability and clarity.

**Weaknesses:**

- The paper introduces various settings for ImplicitSLIM, but it would be beneficial to analyze and summarize the computational complexity of different variants respectively to provide a clearer understanding of their efficiency.
- The paper could benefit from more detailed experimental studies on the influence of hyperparameters. Given the presence of multiple regularization and optimization items in the method, it would be more illustrative to have the model performance w.r.t. different parameter settings.
- Although ImplicitSLIM has shown competitive performance against traditional linear encoders, it shows limitted improvement on deeper models like UltraGCN and RecVAE.

**Questions:**

Please refer to the weaknesses.

---

> ### Author Response · Authors · 2023-11-21
> **Point-to-point response to the weaknesses and questions**
>
> Thank you very much for your review! Please find below a point-to-point response for the weaknesses and questions from the review.
>
> *W1: The paper introduces various settings for ImplicitSLIM, but it would be beneficial to analyze and summarize the computational complexity of different variants respectively to provide a clearer understanding of their efficiency.*
>
> We fully agree with this comment (see also Q2 of Reviewer uknn). In the updated version of the paper, we have added Figure 3 in Appendix E.1 to visualize the tradeoff between computation time, embedding dimensionality, and ranking scores. In particular, they show how much ImplicitSLIM is faster and more memory efficient compared to EASE, and also show how ImplicitSLIM boosts matrix factorization in terms of convergence time as well as in terms of performance.
>
> *W2: The paper could benefit from more detailed experimental studies on the influence of hyperparameters. Given the presence of multiple regularization and optimization items in the method, it would be more illustrative to have the model performance w.r.t. different parameter settings.*
>
> The primary hyperparameter that, in our opinion, indeed deserves extensive experiments is the embedding dimensionality; we have presented comparative studies of different embedding dimensions in Figures 1 and 3 and Table 4. As for the other hyperparameters, there are no significant tradeoffs there, and we just look for their optimal values with Bayesian hyperparameter search.
>
> *W3: Although ImplicitSLIM has shown competitive performance against traditional linear encoders, it shows limitted improvement on deeper models like UltraGCN and RecVAE.*
>
> We note that the application of ImplicitSLIM to all deep models in the experimental section statistically significantly accelerates their performance (except for one very specific case, H+Vamp(Gated) trained on the Netflix dataset and evaluated with Recall@k, in all other cases the speedup is significant).
>
> In addition, we note that the application of ImplicitSLIM to these models improves convergence, with minor computational and memory overhead thanks to relatively low embedding dimensionality in deep models; for example, RecVAE+ImplicitSLIM achieves the best score of RecVAE almost twice faster than RecVAE itself.
>
> Therefore, we believe that applying ImplicitSLIM to deep recommendation models such as RecVAE and H+Vamp may bring advantages other than just improved metrics, and our experiments show that ImplicitSLIM is beneficial for all models where we have been able to apply it.

---

### Official Review · Reviewer_Fj6S · 2023-10-30

**Soundness:** 3 good
**Presentation:** 3 good
**Contribution:** 3 good
**Rating:** 8
**Confidence:** 4

**Summary:**

The paper introduces ImplicitSLIM, an approach leveraging insights from established linear models such as EASE and LLE. The authors advocate its use for initializing and regularizing item and user embeddings across various collaborative filtering architectures. ImplicitSLIM streamlines the process of extracting embeddings, exhibits robust generalization capabilities, and accelerates convergence of the downstream models. As a comparatively lightweight and effective solution, it has the potential to become a valuable tool in representation learning for collaborative filtering, contributing to both theoretical understanding and practical implementation.

The text is well-written and easy to follow. The presentation of the approach is transparent and mathematically sound. The obtained results are convincing and demonstrate the advantages of the proposed solution. I'd vote to accept the paper.

**Strengths:**

- mathematically sound approach with a closed-form solution
- showcases practical efficiency in the standard collaborative filtering task
- good generalization capabilities

**Weaknesses:**

- not a standalone approach which makes training less straightforward
- applicable to embedding-based models only
- the source code is not provided

**Questions:**

There's a promise to provide empirical comparison with more natural-looking regularizer in Appendix E.2. However, not much information is provided there. The promise creates an expectation that there will be a more substantial comparative data with numbers and graphs. Is it planned to be provided?

---

> ### Author Response · Authors · 2023-11-21
> **Point-to-point response to the comments and questions**
>
> Thank you very much for your review and your kind words about our work! Please find below a point-to-point response to the weaknesses and questions from your review.
>
> *W1: not a standalone approach which makes training less straightforward*
>
> Yes, we agree with this statement in the general case: ImplicitSLIM is indeed a modification aimed to improve existing recommender systems rather than a separate model.
>
> However, in order to mitigate this issue as much as possible, we have performed extensive experiments applying ImplicitSLIM to the simplest suitable models, MF and PLRec. Moreover, one of experimental setups (called “ImplicitSLIM init” in the paper) assumes that the item embeddings matrix for matrix factorization is initialized by ImplicitSLIM output, and there are no other computations during the training phase (user embeddings for held-out users are computed during the test phase given item embeddings). This learning procedure is indeed not performed in an end-to-end fashion.
>
> In Section 4.2, we also make an attempt to make the training of ImplicitSLIM more straightforward; there, we show that a model regularized with the SLIM regularizer approximately matches a model regularized with ImplicitSLIM.
>
> *W2: applicable to embedding-based models only*
>
> Yes, we agree that our approach deals only with embedding-based models and is not applicable, e.g., to recommendations based on nearest neighbors. However, most currently used recommender systems employ embedding-based models, and we believe that this does not significantly hinder the applicability and importance of our results.
>
> *W3: the source code is not provided*
>
> By the conference guidelines for anonymity, we indeed have not published the source code openly but it has been attached in the supplementary files here in the submission.
>
> *Q1: There's a promise to provide empirical comparison with more natural-looking regularizer in Appendix E.2. However, not much information is provided there. The promise creates an expectation that there will be a more substantial comparative data with numbers and graphs. Is it planned to be provided?*
>
> We are not certain that we have understood this comment correctly (and we apologize if not) but perhaps we have not been clear enough regarding what Appendix E.2 sets out to do. Appendix E.2 presents several simplifications to the default theoretical ImplicitSLIM definition that can significantly simplify the training process in practice (approximations a-c). All of these simplifications have succeeded in the sense that they speed up training while having no effect on the performance metrics. Therefore, the ImplicitSLIM setups evaluated in the paper already incorporate these simplifications, and the comparative data would just contain the same performance metrics over and over.

---

### Official Review · Reviewer_EMBd · 2023-10-31

**Soundness:** 2 fair
**Presentation:** 2 fair
**Contribution:** 2 fair
**Rating:** 3
**Confidence:** 1

**Summary:**

This paper proposes a novel unsupervised learning approach for sparse high-dimensional data. The method learns local structure of data (embeddings) in the embedding space where the embeddings of similar objects to be similar.

**Strengths:**

This paper learns embeddings with closed form solutions.
Good Experimental study.

**Weaknesses:**

- This paper is very theoretical and hard to follow the formulas.

**Questions:**

This paper is too theoretical than any other submissions on NeurIPS and ICLR. I hardly follow the content.

---

### Official Review · Reviewer_uknn · 2023-11-02

**Soundness:** 3 good
**Presentation:** 3 good
**Contribution:** 2 fair
**Rating:** 3
**Confidence:** 4

**Summary:**

This paper proposed a method named implictSLIM that can be integrated with other embedding-based methods.

**Strengths:**

Strengths:

-	The approaches addresses the memory-intensive and scalability issues of SLIM-like models in collaborative filtering.
-	The authors provided various experiments on publicly available benchmark datasets
-	Source code is also provided
-	Many appendices were given for more explanation

**Weaknesses:**

Weaknesses:

-	In Section 3 Proposed Approach, the authors should explain more on the formular choices when developing ImplicitSLIM. For example, why do we use LLE, but use the neighbourhood of NN(i) = {1,2,…,I} \ {i} to make it ‘global’ (Section 3.1)? Why do we drop the sum-to-one constraint (Section 3.2) (the authors did mention they have no good reasons, but why)? In my opinion, Section 3 is very important and the authors should provide deeper explanations and discussions about this. Otherwise it’s very hard to convince the readers.
-	In the first paragraph after Figure 2, the authors mentioned that “In addition, ALS applied to MF regularized by …. about 5x faster”, I may missed it but where do we find the ‘5x faster’ comparison?
-	In Figure 1, why ImplicitSLIM init + SLIM reg and SLIM-LLE init cannot have results with > 500 embedding dimension? Could we please add an appendix on the ‘high computational costs’?
-	In Section 4.1, in the last sentence, the authors mentioned that “this procedure may be less stable … fewer calls to ImplicitSLIM”. Why is that?
-	In Table 1, the performance results are not really impressive. For example, in the Appendix E.1, the authors mentioned that “Moreover, ImplicitSLIM is also faster than EASE… could replace EASE in some cases due to lower computational time and comparable performance.”. Please explain which cases?
-	In Appendix E.3, Table 5, why RecVAE + ImplicitSLIM cannot perform better than RecVAE?
-	References should be sorted as the current version is hard to follow (minor)

Overall, more works need to be done.

**Questions:**

Please refer to the above comments.

---

> ### Author Response · Authors · 2023-11-21
> **Point-to-point response to the questions and weaknesses**
>
> Thank you very much for your review and comments! Please find a point-to-point response below.
>
> *Q1: In Section 3, the authors should explain more on the formular choices when developing ImplicitSLIM. For example, why do we use LLE, but use the neighbourhood of NN(i) = {1,2,…,I} \ {i} to make it ‘global’ (Section 3.1)? Why do we drop the sum-to-one constraint (Section 3.2)...*
>
> Thank you for this remark – we agree that this could be explained better. First, the chosen neighborhood function and dropping the sum-to-one constraint are both necessary to perform the trick in Section 3.2 that makes it possible to avoid inverting huge matrices.
>
> Considering EASE as a special case of the first step of LLE gives us additional motivation to use this neighborhood function, as we mention in Section 3.1; in general, the similarity of EASE and LLE motivates us to use LLE as a model for item/user representations learning.
>
> The sum-to-one constraint at the first step implicitly ensures the sum-to-zero constraint at the second step of the original LLE, but in this work we propose a different second step, which is defined as an unconstrained optimization problem, so we have a technical reason to drop the sum-to-one constraint. We have updated Sections 3.1 and 3.2 to explain this more clearly.
>
> *Q2: ...“In addition, ALS applied to MF regularized by …. about 5x faster”, I may missed it but where do we find the ‘5x faster’ comparison?*
>
> Thank you for noticing this! In the updated version of the paper, we have added Fig. 3 in Appendix E.1 to visualize the tradeoff between computation time, embedding dimensionality, and ranking scores. In particular, it shows how much ImplicitSLIM is faster and more memory-efficient compared to EASE, and also shows how ImplicitSLIM boosts matrix factorization in terms of convergence time as well as in terms of performance.
>
> *Q3: In Figure 1, why ImplicitSLIM init + SLIM reg and SLIM-LLE init cannot have results with > 500 embedding dimension? Could we please add an appendix on the ‘high computational costs’?*
>
> Both of these setups, unlike the others, require huge matrix inversions. Moreover, SLIM-LLE init setup requires the computation of eigenvectors for the smallest eigenvalues, and the ImplicitSLIM init + SLIM reg setup could not be solved with ALS, so we had to employ gradient descent for this problem. As a result, computational costs for these two settings are much higher, especially when the dimensionality of embeddings is large; hence we have evaluated these setups with embedding dimensions equal to 500 and lower. In the updated version of the paper, we have expanded Appendix C to explain it.
>
> *Q4: In Section 4.1, in the last sentence, the authors mentioned that “this procedure may be less stable … fewer calls to ImplicitSLIM”. Why is that?*
>
> The instability that may arise during the application of ImplicitSLIM to downstream models may lead to a sudden drop in performance metrics while training, followed by a recovery after several training epochs. For this reason, we update the embeddings using ImplicitSLIM once in a constant number of epochs to allow the model to recover. This effect can be observed in Figure 2 (convergence plots).
>
> *Q5: In Table 1, the performance results are not really impressive... “Moreover, ImplicitSLIM is also faster than EASE… could replace EASE in some cases due to lower computational time and comparable performance.”. Please explain which cases?*
>
> Please see the answer to Q2 above: we have added Figure 3 in Appendix E.1 (and a corresponding explanation) to visualize how ImplicitSLIM outperforms EASE in terms of wall clock time and memory at the cost of a slight performance drop.
>
> *Q6: In Appendix E.3, Table 5, why RecVAE + ImplicitSLIM cannot perform better than RecVAE?*
>
> In this experiment, we first took RecVAE+ImplicitSLIM fitted to maximize the overall score on the current dataset, and then evaluated it separately for high, mid, and low popularity items. The model was not fitted to maximize scores that correspond to high, mid, and low popularity items individually.
>
> This experiment shows that the application of ImplicitSLIM to RecVAE improves its overall performance on the MovieLens20M dataset by improving the embeddings of highly popular items, while embeddings of mid- and low-popularity items may even slightly degrade. This observation could be explained by the fact that, as we note in Appendix D.1, we zero out columns in matrix A that correspond to unpopular items, which empirically leads to better overall performance but may hurt the performance for unpopular items. We have added an explanation of this effect to Appendix E.3.
>
> *Q7: References should be sorted as the current version is hard to follow (minor)*
>
> We apologize but we didn’t understand this remark: the references are sorted alphabetically by the first author’s last name; we have checked last year’s ICLR papers and saw the references sorted in the same way…

---

### Meta-Review · Area_Chair_3Vov · 2023-12-11

**Metareview:**

This paper proposes ImplicitSLIM, an unsupervised learning approach for sparse high-dimensional data, specifically applied to collaborative filtering. ImplicitSLIM enhances embedding-based models by efficiently extracting embeddings from SLIM-like models, leading to improved performance and faster convergence for both modern and traditional collaborative filtering methods.

Overall, I think this paper studies an important representation learning problem, i.e., the representation of high-dim data, which would benefit the broad data science techs, such as the tabular learning, recommendation, social networks etc. The proposed ImplicitSLIM method improves embedding-based models by extracting embeddings from SLIM-like models in a computationally cheap and memory-efficient way, without explicit learning of heavy SLIM-like models. The theoretic analysis, experiment results and the presentation are all satisfactory. Although there is variance among the reviews' scores, after carefully reading this paper, review, and rebuttal discussions, I would like to recommend accepting this paper.

**Justification For Why Not Higher Score:**

There are two reviews with negative ratings, with some concerns on the readability and technical details.

**Justification For Why Not Lower Score:**

Overall, I think this paper studies an important representation learning problem, i.e., the representation of high-dim data, which would benefit the broad data science techs, such as the tabular learning, recommendation, social networks etc. The proposed ImplicitSLIM method improves embedding-based models by extracting embeddings from SLIM-like models in a computationally cheap and memory-efficient way, without explicit learning of heavy SLIM-like models. The theoretic analysis, experiment results and the presentation are all satisfactory. Although there is variance among the reviews' scores, after carefully reading this paper, review, and rebuttal discussions, I would like to recommend accepting this paper.

---

### Decision · Program_Chairs · 2024-01-16

Accept (poster)